# Heterologous Booster Immunization Based on Inactivated SARS-CoV-2 Vaccine Enhances Humoral Immunity and Promotes BCR Repertoire Development

**DOI:** 10.3390/vaccines12020120

**Published:** 2024-01-24

**Authors:** Xinghang Li, Fengyuan Zeng, Rong Yue, Danjing Ma, Ziyan Meng, Qi Li, Zhenxiao Zhang, Haobo Zhang, Yuansheng Liao, Yun Liao, Guorun Jiang, Heng Zhao, Li Yu, Dandan Li, Ying Zhang, Longding Liu, Qihan Li

**Affiliations:** Yunnan Key Laboratory of Vaccine Research and Development on Severe Infectious Diseases, Institute of Medical Biology, Chinese Academy of Medical Sciences & Peking Union Medical College, Kunming 650118, Chinazhangy@imbcams.com.cn (Y.Z.)

**Keywords:** SARS-CoV-2, vaccine, sequential immunization, B-cell receptor, humoral immunity

## Abstract

Recent studies have indicated that sequentially administering SARS-CoV-2 vaccines can result in increased antibody and cellular immune responses. In this study, we compared homologous and heterologous immunization strategies following two doses of inactivated vaccines in a mouse model. Our research demonstrates that heterologous sequential immunization resulted in more immune responses displayed in the lymph node germinal center, which induced a greater number of antibody-secreting cells (ASCs), resulting in enhanced humoral and cellular immune responses and increased cross-protection against five variant strains. In further single B-cell analysis, the above findings were supported by the presence of unique B-cell receptor (BCR) repertoires and diversity in CDR3 sequence profiles elicited by a heterologous booster immunization strategy.

## 1. Introduction

COVID-19 is a serious public health threat caused by the coronavirus SARS-CoV-2, resulting in significant economic losses. As of September 2023, over 770 million individuals have contracted the virus, leading to the loss of 6.9 million lives [1]. The virus primarily infects human respiratory epithelial cells, causing various respiratory and systemic symptoms [2]. B-cells play a vital role in adaptive immunity by generating neutralizing antibodies that defend against SARS-CoV-2 infection and prevent reinfection [3,4,5]. Furthermore, specific monoclonal antibodies or convalescent plasma with high potency may play a distinct role in controlling COVID-19 [6]. Although new variants of SARS-CoV-2 have emerged [7,8,9,10], vaccinated individuals possess antigen-specific B-cells capable of neutralizing not only the vaccine strain itself but also multiple variants, thereby reducing the rate of severe illness and death, especially among elderly individuals and specific populations [11,12,13,14].

The ongoing prevalence of COVID-19 and the emergence of new variants have led to an increase in the prevalence of breakthrough infections, despite the administration of booster shots [15]. However, recent studies have shown that a third dose of the COVID-19 vaccine can effectively reactivate the immune response to SARS-CoV-2, resulting in decreased rates of hospitalization and readmission [16,17,18]. While certain immune markers improve after the third vaccine dose, importantly, repetitively administering the same vaccine may not yield optimal outcomes, as the body’s immunity has limits in its response to a specific vaccine. This could be attributed to the fact that the body’s immunity has an upper limit for antibody production in response to a specific vaccine; hence, an infinite increase in antibodies cannot be achieved [19] However, studies on sequential immunization have found that specific combinations can generate higher levels of humoral and cellular immunity, providing enhanced protection [20,21,22].

Sequential immunization involves administering separate technical lines of vaccine for prime and booster immunization. This approach was proposed due to vaccine supply shortages and rare safety concerns, such as vaccine-induced immune thrombotic thrombocytopenia (VITT) caused by adenovirus vector vaccines. Those who have already received this type of vaccine must consider sequential immunization to be a necessary precaution [23,24]. Studies have shown that certain sequential vaccination approaches may provide more effective defense against SARS-CoV-2 than homologous booster immunizations, such as primary immunization using an adenovirus vector vaccine and subsequent booster immunization with an mRNA vaccine that can yield well-rounded and resilient humoral and cellular immunity [25,26,27]. However, previous studies on sequential immunity have predominantly focused on analyzing antibody and cellular immunity levels at the population level, with few examining the differences between sequential immunity and homologous immunity within a single B-cell lineage. Zhao and colleagues [28] analyzed B-cell receptor profiles in mice following sequential immunization, thereby identifying notable differences induced by these vaccination strategies.

Most individuals in China received complete immunization with two doses of inactivated vaccine between 2021 and 2022. Thus, understanding heterologous immunization effects in this population is important for facing future COVID outbreaks. In this study, we evaluated the immunogenicity of booster recombinant protein vaccines, mRNA vaccines, and inactivated vaccines in mice after two doses of inactivated vaccines. In addition, we would like to emphasize the comparison of BCR profiles induced by different immunization strategies. In this paper, we acquired B-cell receptor (BCR) profiles at the single B-cell level and compared the frequency of germline gene expression and CDR3 sequences to analyze the crucial role of the B-cell receptor (BCR) pool in defending against SARS-CoV-2.

## 2. Materials and Methods

### 2.1. Mouse Immunization and Ethics

Six- to eight-week-old female BALB/c mice were purchased from Charles River (Beijing, China) with license no. SCXK (jing) 2021-0006. The mice were randomly divided into four groups of five. The inactivated vaccine (GenBank No: MT226610.1) was sourced from the Chinese Academy of Medical Sciences (CAMS), Institute of Medical Biology (IMB), and the full-length S protein (Cat: 40589-V08H4) of SARS-CoV-2 (Wuhan-Hu-1 strain) was procured from SinoBiological (Beijing, China). The mRNA vaccines consist of mRNA that encodes the full-length S protein of SARS-CoV-2 (Wuhan-Hu-1 strain), encapsulated in lipopolyplex (LPP) with a core-shell structure. The mRNA vaccine was a gift from Stemirna Co., Ltd. (Shanghai, China). The Addavax adjuvant was obtained from InvivoGen (San Diego, CA, USA). Each group of mice received two doses of inactivated vaccine (1/5 of a human dose, 30 U) at a 14-day interval. The mice in group B received an additional dose of the inactivated vaccine 14 days after the prime immunization. The mice in group C were injected with a mixture of S-protein (10 μg/100 μL) and Addavax. The mice in group D were injected with 10 μg of mRNA vaccine. Finally, the mice in group A were sacrificed 14 days after receiving two doses of the inactivated vaccine, and their blood and spleens were collected. The remaining three groups were euthanized 14 days after the administration of the third dose of vaccine (see Figure 1a). All animal experiments were designed and performed according to the principles of the “Guide for the Care and Use of Laboratory Animals” and the “Guidance for Experimental Animal Welfare and Ethical Treatment”. The protocols were approved by the Experimental Animal Management Association of IMB, CAMs (DWSP 202003 005).

### 2.2. ELISA

The SARS-CoV-2 (Wuhan-Hu-1 strain) S1 protein (Sanyou Biopharmaceuticals Co., Ltd., Shanghai, China) was added at a concentration of 0.1 μg/100 μL/well to 96-well ELISA plates (Corning, NY, USA). Subsequently, serum was added to the wells using 2-fold multiplicative dilution and then incubated at 37 °C for 1 h. Following this, the wells underwent five washes using PBST before 100 μL of horseradish peroxidase-conjugated goat anti-mouse IgG (Thermo Fisher, Waltham, MA, USA) was applied to each well and incubated again at 37 °C for a duration of 40 min. During the serum IgG subtype detection experiments, the antibodies were replaced with horseradish peroxidase-conjugated goat anti-mouse IgG1 and horseradish peroxidase-conjugated goat anti-mouse IgG2a (Abcam, Cambridge, MA, USA). The plate was then washed five times with PBST buffer. Then, 100 μL of TMB substrate (3,3′,5,5′-tetramethylbenzidine) was added for 10 min to allow for color development. Finally, 50 μL of ELISA stop solution (Solarbio, Beijing, China) was added to terminate color development. The results were read using an ELISA plate reader (Gene Company, Hong Kong, China) at a wavelength of 450 nm. The terminal dilution criterion was met when the OD value in the sample wells exceeded 2.1 times that of the blank wells.

### 2.3. ELISpot Assay

First, spleens were collected from mice, and splenocytes were prepared into single cell suspensions in 6-well plates (CellPro, Suzhou, China) containing RPMI 1640 medium (Gibco, Grand Island, NY, USA). These suspensions were then added to 15 mL centrifuge tubes, and mouse lymphocyte isolates (Dakewe Biotech Co., Ltd., Shenzhen, China) were obtained according to the instructions for mouse lymphocyte isolates. The mouse IFN-γ ELISPOTPLUS kit (ALP) and mouse IL-4 ELISPOTPLUS kit (ALP) (Mabtech, Nacka Strand, Sweden) were utilized following the manufacturer’s instructions. Approximately 5 × 10^5^ splenic lymphocytes were added to the wells and cultured in a medium with a final concentration of 2 μg/mL S1 protein. Ten microliters of PHA (Dakewe Biotech Co., Ltd., Shenzhen, China) was added to the positive control wells, while the negative control wells contained cells and RIPA 1640 medium only. The plates were incubated in an incubator containing 5% CO_2_ at 37 °C for 24 to 36 h. Post incubation, the spots were color developed as per the instructions provided and then read with an ELISpot reader (CTL, Shaker Heights, OH, USA).

### 2.4. Antibody-Secreting Cell ELISpot

The instructions for ELISpot Flex: Mouse IgG (ALP) (Mabtech, Nacka Strand, Sweden) were followed. Briefly, PVDF membranes in the wells of ELISpot plates (Merck Millipore, Billerica, MA, USA) were first activated with 20 μL of sterile 35% (*v*/*v*) ethanol solution on the first night, followed by five washes with sterile PBS solution. After adding SARS-CoV-2 S1 protein at a concentration of 5 μg/mL overnight, the membrane was washed with PBS solution five times. Next, the membrane was blocked with RPMI 1640 medium containing 10% FBS for a minimum of 30 min. Next, 5 × 10^5^ mouse splenic lymphocytes were added to each well containing RPMI 1640 medium containing 1% penicillin and incubated at 37 °C in 5% CO_2_ for 24 h. Subsequently, the membrane was incubated at 37 °C. After being incubated for 24 h, primary and secondary antibodies were added and incubated separately, following the provided instructions. The ELISpot reader was used to read the spots after the color was developed with the BCIP/NTB color solution for 20 min.

### 2.5. Pseudovirus Neutralization Assay

Replication-deficient recombinant SARS-CoV-2 pseudovirus (rVSV-SARS-CoV-2) was kindly donated by the Wang laboratory. The experimental manipulation was performed as previously described [29]. Briefly, the samples to be tested were diluted at a 2-fold multiplicity in 100 μL of 90% DMEM (Servicebio, Wuhan, China), 10% fetal bovine serum (FBS) (Biochannel, Nanjing, China), and 1% penicillin (SolarBio, Beijing, China) medium, followed by the addition of 50 μL of rVSV-SARS-CoV-2 and incubation at 37 °C for 1 h. Subsequently, 100 μL 4–5 × 10^5^/mL 293T-hACE2 cells was added. The cells were then incubated at 37 °C in 5% CO_2_ for 24 h. Brotelite Plus (PerkinElmer, Waltham, MA, USA) was utilized for color development by following the manufacturer’s instructions, and fluorescence intensity values were analyzed using Synergy 4 (BioTek, Winooski, VT, USA).

### 2.6. Sorting of Antigen-Specific B-Cells and BCR Gene Amplification

Splenic lymphocyte suspensions from each group of mice were pooled and resuspended in 300 μL of FACS buffer, which is a PBS solution containing 2 mM EDTA and 2% FBS. Prior to cell staining, the cells were preincubated with streptavidin-PE, and 2 μg/10^7^ cells were biotinylated with Avi-tagged SARS-CoV-2 antigens (SinoBiological, Beijing, China) at a 4:1 molar ratio for 30 min. Subsequently, we added 2 μg/10^7^ cells of anti-mouse B220-FITC and 1 μg/10^7^ cells of goat anti-mouse IgG-APC to the cell suspension and incubated for 30 min away from light. The desired cells were then sorted using flow cytometry (Backman, Miami, FL, USA) and placed in 96-well PCR plates (Yeasen, Shanghai, China). All of the above antibodies were purchased from Biolegend (San Diego, CA, USA).

The steps for Ig gene amplification of antigen-specific B-cells are as follows. A mix containing 2 μL 5× PrimeScript IV cDNA Synthesis Mix (Takara, Japan), 2 μL deoxynucleotide triphosphates (Takara, Japan), 0.5 μL RNA inhibitor (Takara, Japan), 0.1 μL IGEPAL CA-630 (Beyotime, Shanghai, China), 1.25 μL 0.1 M DTT (1,4-Dithio-DL-threitol; Threo-1,4-dimercapto-2,3-butandiol) (Solarbio, Beijing, China), and 11.15 μL RNase-free Water (Takara, Tokyo, Japan) added to each well of a single cell-sorted 96-well plate. Reverse transcription was performed using a Bio-Rad PCR instrument (Hercules, CA, USA) with temperature cycles set to 30 °C for 10 min, 42 °C for 20 min, and 70 °C for 15 min. The resulting cDNA was stored at −20 °C. Subsequent nested PCR of heavy and light chain genes was performed according to previously reported steps [30,31]. The PCR products were separated on a 1% agarose gel, and those with correct heavy and light chain bands were then used for Gibson ligation (Vazyme, Nanjing, China) cloning and insertion into human IgG expression vectors (AbVec2.0-IGHC and AbVec1.1-IGKC) and transformation into competent cells (Sangon, Shanghai, China). The next day, the plasmids were extracted from monoclonal bacterial fluids and confirmed via Sanger sequencing (Sangon, Shanghai, China). Using the IMGT/V-QUEST method (https://www.ncbi.nlm.nih.gov/projects/igblast/ (acessed on 12 March 2023)), the appropriate IgV, IgD, and IgJ germline genes compatible with the mice were identified by searching the database for correctly sized heavy and light sequences.

### 2.7. Flow Cytometry

Mice were euthanized 2 weeks after completion of their respective immunization schedules. The popliteal lymph nodes were then collected and mechanically ground to obtain single cell suspensions. These suspensions were centrifuged at 1000× *g* for 10 min. Afterward, the cells were resuspended in 100 μL of FACS buffer and stained with appropriate flow cytometry antibodies for 30 min. Subsequently, lymph node germinal centers, including GC B (B220+ GL7+ CD95+) and TfH (CD3+ CD4+ CXCR5+ PD-1+) cells, were identified using flow cytometry (BD Biosciences, San Jose, CA, USA). The data were analyzed using FlowJo software (Version X).

### 2.8. Serum Multiplex Cytokine Analysis

Ten serum cytokines (IFN-γ, IL-1β, IL-2, IL-4, IL-5, IL-6, IL-10, IL-12p70, CXCL-1, and TNF-α) were assayed by the V-PLEX Proinflammatory Panel 1 Mouse Kit (Meso Scale Discovery, Rockville, MD, USA). The assay was performed by adding serum samples and diluted MSD calibrators and controls to the microplate separately for incubation and subsequent detection by electrochemiluminescence. The results were analyzed using a QuickPlex SQ 120 instrument (MSD, MD, USA) and DISCOVERY WORKBENCH^®^ 4.0 software.

### 2.9. Statistical Analysis

All statistical analyses were conducted using one-way ANOVA (and nonparametric or mixed tests) using GraphPad Prism (GraphPad software Version 9.0). Data were defined as statistically significant if the *p* value was <0.05.

## 3. Results

### 3.1. Booster Immunization Significantly Increases Antigen-Specific Antibodies in Mice

After receiving two doses of the inactivated vaccine (Group A), the geometric mean titer (GMT) of IgG antibodies against the S1 protein in mice was 9701. In contrast, the level of IgG antibodies increased to 33,779 in the homologous booster immunization group (Group B). Furthermore, in the heterologous booster protein vaccine (Group C) and mRNA vaccine (Group D), the levels of IgG antibodies reached 178,289 and 135,118, respectively. These levels were 13.9–18 times higher than those of group B and 4–5.7 times higher than those of group B (Figure 1b). The antibodies levels in groups C and D showed a significant increase compared to groups A and B (*p* < 0.05).

Moreover, we evaluated the levels of pseudovirus neutralizing antibodies against the wild-type (WT, Wuhan-Hu-1) strain and five variants, including Alpha (B.1.1.7), Beta (B.1.351), Gamma (P.1), Delta (B.1.617.2), and Omicron (BA.1). First, the GMT for neutralizing antibodies against the WT strain of pseudovirus was obtained for the four groups, with values of 4850, 33,802, 22,286, and 1213. The booster immunization group showed a significant enhancement in neutralizing antibody levels, with a 4–27.8-fold increase compared to group A, which did not receive a third dose of immunization (*p* < 0.05). There was no significant decrease in the levels of antibodies against the Alpha, Beta, and Gamma variants observed in groups C and D when challenged with the five variant strains. However, groups A and B exhibited a decrease in antibody levels. For instance, the GMTs of group A in the presence of each of the three mutants were 2786, 1838, and 2425, respectively, which were 0.38–0.57-fold higher than the levels of antibodies against the WT strains. When challenged with the Delta and Omicron BA.1 variant strains, the GMTs of all four groups significantly decreased. For instance, when challenged with the Omicron BA.1 variant strain, the GMTs of the four groups were 348.2, 2786, 1838, and 174.1, respectively. There remained a difference of 2- to 16-fold in the antibody titers in the booster immunization group compared to those of group A. Antibody titers from heterologous booster immunization were 5.3- to 8-fold greater than those from homologous booster immunization.

We measured the levels of S1-specific IgG antibodies 90 days after immunization to assess immune persistence. The four groups exhibited varying degrees of antibody decline. Group A demonstrated a decrease in antibody GMT levels to 3200, while groups B, C, and D showed decreases to 16,890, 25,600, and 29,407, respectively. Although there was a higher multiplicative decrease in antibody levels in the heterologous immunization group compared to the other two groups, there remained a significant discrepancy in antibody levels between the heterologous immunization group and the group that only received two doses of inactivated vaccine (Figure 1i).

### 3.2. Heterologous Booster Immunization Induces Stronger Antibody-Secreting Cells and Induces IgG Antibody Class Switching

The mean number of antibody-secreting cells induced across the four groups ranged from 69.9 to 454.5 per 5 × 10^5^ cells. Group D, which received the heterologous booster mRNA vaccine, displayed the highest numbers, and group A, which received two doses of the inactivated vaccine, displayed the lowest. All three groups receiving boosters (B, C, and D) showed significant differences (*p* < 0.05) from group A (Figure 2a,b). We compared IgG typing of antibodies induced using different immunization strategies. The predominant IgG isotype present in all four groups was IgG2a, with respective IgG1/IgG2a ratios of 0.33, 0.39, 0.74, and 0.56 (Figure 2c). Although there was little difference in the ratios between group A and group B, groups C and D displayed a significant increase in IgG1/IgG2a ratios, with group C having the highest ratio among the four groups.

Because B-cell development and antibody class switching primarily occur in the germinal centers of the lymph nodes, we utilized flow cytometry on single cell suspensions from mouse lymph nodes. Our results demonstrated a significant increase in GC B-cells (B220+ CD95+ GL7+) in the lymph nodes of immunized mice compared to those of unimmunized mice. Among the groups, group C exhibited the highest ratio of GC B cells, with a mean of 12.62% B220+ cells. In contrast, group A had the lowest percentage of GC B cells, at 4.23% (Figure 2d). Groups D and B had 7.83% and 7.03%, respectively. In terms of TfH-cells (CD4+ CXCR5+ PD-1+), groups C and D showed higher values than groups A and B, with group D having the highest mean value of 0.65% (Figure 2e).

### 3.3. A Heterologous Boost Elicits Robust Cellular Immune Responses

ELISpot assays were conducted in this study to measure the abundance of IL-4- and IFN-γ-secreting lymphocytes from splenic lymphocytes of mice 14 days post immunization. Regardless of IL-4 and IFN-γ expression, there was a consistent trend in the number of spots presented, with the highest number of spots observed in the group receiving the heterologous booster of recombinant protein vaccine (group C), followed by the group receiving the heterologous booster of mRNA vaccine (group D), then the group receiving the 3-dose homologous immunization (group B), and the lowest number of spots observed in the group receiving the two doses of the inactivated vaccine (group A). The average number of induced IFN-γ+ PBMCs in the four groups was 20.0, 38.4, 61.2, and 49.6 per 2.5 × 10^5^ cells, respectively. The heterologous boost groups (groups C and D) exhibited significant differences (*p* < 0.05) compared to the two-dose inactivated vaccine group (group A). The three-dose inactivated vaccine group (group B) had a higher number of induced IFN-γ+ PBMCs than the two-dose inactivated vaccine group (group A), but there was no significant difference (*p* > 0.05) (Figure 2f–h). The average number of induced IL-4+ PBMCs was 48.4, 82.6, 284, and 251.6 per 5 × 10^5^ cells, respectively. Groups C and D significantly differed from groups A and B. Group B showed a marginally insignificant difference compared to group A. Additionally, the results obtained through flow cytometry demonstrated a slight increase in the number of CD8+ T-cells in the splenic lymphocytes of the mice in groups C and D in comparison to those in groups A and B. However, there was no statistically significant difference (*p* > 0.05) (Figure 2j).

### 3.4. Booster Immunization Affects the Levels of Cytokines in Serum

We quantified ten cytokines (IFN-γ, TNF-α, IL-1β, IL-2, IL-4, IL-5, IL-6, IL-10, IL-12, and CXCL-1) in the serum of immunized mice (Figure 3). Different immunization strategies were found to result in differences in serum levels of some of these cytokines. Among these cytokines we examined, the levels of most of them showed an increasing trend in the booster immunization group. In group B, there was an increase in the levels of IL-2, IL-5, IFN-γ, and IL-6. Group C showed elevated levels of IL-4, IL-5, IL-6, and IL-10, with IL-4 and IL-5 demonstrating a significant increase. Group D exhibited elevated levels of IL-2, IL-5, IL-10, IFN-γ, and IL-6 to a certain extent. Overall, cytokine levels remained stable 14 d after the completion of the immunization procedures for the different strategies, and no further abnormal cytokine levels were observed.

### 3.5. Homologous and Heterologous Boosters Promotes Continued Development of the BCR Lineage

#### 3.5.1. Homologous and Heterologous Boosters Both Contribute to the Continuous Maturation of Germline Genes after the Prime Immunization

Antigen-specific B-cells were isolated via flow cytometry and subsequently subjected to nested PCR and agarose electrophoresis (Figure 4a). Following confirmation of the correct size of the band of interest, it was cloned and inserted into the expression vector for Sanger sequencing. The resulting sequences were compared to the database to obtain the needed Ig gene sequence and lineage information. Simultaneously, the ratio of somatic high-frequency mutations (SHM) was determined by aligning the obtained sequences in the database. The heavy chain exhibited an average SHM ratio of 3.22%, 2.91%, 2.73%, and 2.94% in groups A, B, C, and D, respectively. The average mutation rates in the light chains were 1.63%, 1.17%, 1.58%, and 1.27%, respectively (Figure 4b,c).

The germline genes that encode the V fragment of IgH are predominantly IGHV1, IGHV3, IGHV2, IGHV5, and IGHV14. A smaller quantity of its component is also encoded by IGHV10, IGHV4, IGHV6, IGHV7, IGHV12, and IGHV19 (Figure 4d). The distribution of IgH J fragments showed similarity between IGHJ2 and IGHJ3 in all four groups. However, the proportion of IGHJ1 expression in groups B and C was approximately 20% higher than that in the other two groups. On the other hand, the proportion of IGHJ4 expression was lower in groups B and C than in the other two groups (Figure 4f). For IgK, the V fragment is primarily encoded by germline genes located on IGKV4, IGKV1, and IGKV3. Additionally, there is some distribution on IGKV6, IGKV8, IGKV9, IGKV10, IGKV12, and IGKV14 (Figure 4e). The frequency of use among the five germline genes in groups A and B was relatively similar in terms of IGKJ expression. Among these, all four groups showed low use of IGKJ3 and IGKJ4. Furthermore, group D exhibited a significantly lower frequency of IGKJ5 expression than the other three groups. Group C had a lower frequency of IGKJ1 expression than the remaining three groups, while groups C and D showed a higher utilization of IGKJ2 than groups A and B (Figure 4g). The four groups had similar characteristics in the distribution of germline genes. Especially in the V region, their respective high expression frequency germline genes are similar. However, they also have some direct frequency differences, such as the more obvious variations in IGHJ1 and IGHJ4.

Subsequently, a closer examination of the subtypes of germline genes with high expression frequencies within the four groups examined in this study uncovered the following characteristics. Some of the high-frequency germline genes found in the two-dose inactivated vaccine group (Group A) were also present in the other three groups, including IGHV1-77, IGHV3-8, and IGHV1-9 for IgH, as well as IGKV-4-59, IGKV-4-53, IGKV-9-120, and IGKV-10-96 for IgK (Figure 4h,j). The corresponding subclasses of these germline genes were also prevalently expressed in each group, although the frequency of their occurrence varied. For instance, IGHV3-8*02 had the highest rate of occurrence, at 21% in group C. In groups D and A, the rates of occurrence were 7.05% and 9.55%, respectively. Another example is IGHV1S34*01. It exhibited a frequency of 7.27% in group A, whereas in groups B, C, and D, the frequency dropped to 1.75%, 1.69%, and 0%, respectively. This implies that the third booster immunization resulted in the expansion and reduction in the expression of specific germline genes. Additionally, the group that received the heterologous booster mRNA vaccine had a more uniform distribution of expression of heavy and light chain germline genes compared to the other three groups. Furthermore, the frequency of germline genes was no longer excessively high.

Similar phenomena were observed in the pairing of germline genes V-J, where expression of high-frequency genes in group A remained prevalent in the other three groups. An instance is the IGHV1-77-IGKJ-02 gene of IgH, with utilization rates of 3.51%, 6.10%, and 7.05% in groups B, C, and D, respectively (Appendix A). It ranked fourth, third, and first for utilization rate in each group, respectively. IgK IGKV4-59-IGKJ5 expression appeared in the remaining three groups with frequencies of 1.72%, 4.55%, and 5.97%, respectively, ranking sixth, third, and first in this group. There were also genes with a lower frequency of expression in group A but a higher prevalence in the other three groups, such as IGHV1-14-IGHJ1 and IGHV3-08-IGHJ2 in IgH, and IGKV3-4-IGKJ1 and IGKV9-120-IGKJ2 in IgK (Appendix A). After the third booster immunization, certain V-J gene pairs appeared or vanished: for example, IGHV3-8-IGHJ2, predominantly in group C. This combination was infrequent or nonexistent in the other groups in this form.

#### 3.5.2. Heterologous Boosting Increases the Diversity of CDR3

The total number of IgH and IgK sequences acquired for each of the four groups, along with their respective H-CDR3 and K-CDR3 species counts, are displayed in Table 1.

The CDR3 species/sequence ratios for the heavy chain were 0.54, 0.54, 0.55, and 0.79, while the ratios for the light chain were 0.32, 0.45, 0.44, and 0.57 for each of the four groups. Notably, all four groups exhibited higher ratios for IgH than IgK, demonstrating differences ranging from 0.08–0.22. This indicates a greater diversity in H-CDR3 relative to K-CDR3. Additionally, it was discovered that in group D, the ratios of IgH and IgK were the highest among the four groups, reaching 0.79 and 0.57, respectively. Conversely, the ratio of IgK in group A was only 0.32, which was obviously lower than that in the remaining three groups.

Since all booster immunization groups, B, C, and D, were compared to group A immunized with two doses of inactivated vaccine, we aimed to compare the number of identical CDR3 sequences in groups B, C, and D with those in group A. Plotting the crossover of the emerging H-CDR3 and K-CDR3 sequences of the four groups on a Venn diagram (Figure 5a,b), groups B, C, and D each shared 20, 35, and 15 identical H-CDR3s with group A. These shared sequences accounted for 32.8%, 21.5%, and 12.3% of the respective total number of H-CDR3 sequences in each group. There were 58, 55, and 78 identical K-CDR3 sequences, representing 44.0%, 38.2%, and 33.8% of the total K-CDR3 sequences in groups B, C, and D, respectively. In summary, both the H-CDR3 and K-CDR3 groups exhibited varying ratios of CDR3 species/sequence. Of the three groups, the lowest ratio was observed among group D, which received the heterologous boost of mRNA vaccine. Group C, which received a heterologous boost of protein vaccine, exhibited a slightly higher ratio, while group B, which received a homologous boost of inactivated vaccine, demonstrated the highest ratio.

Additionally, when comparing the length of CDR3, it was found that the median length of H-CDR3 among the four groups was 12, 12, 12, and 13 amino acids, respectively, with mean lengths of 11.86, 11.98, 11.98, and 12.28 amino acids (Figure 5c). The median K-CDR3 length in each of the four groups was 9 amino acids, with average lengths of 8.97, 8.97, 9.01 and 8.99 amino acids (Figure 5d). The median and mean lengths of the CDR3 regions were comparable across the four groups, and their direct length distributions showed no significant variation (*p* < 0.05).

## 4. Discussion

Traditionally, certain vaccines require multiple doses to ensure effective protection. For instance, the vaccine for pertussis, diphtheria, and tetanus (DTap) is recommended to be administered to children four to five times between the ages of three months and six years. The frequent dosages are typically necessary to achieve lasting immunity [32]. Even if an individual has completed their childhood immunization program, certain vaccines may still require booster immunization in adulthood, including the hepatitis B vaccine. It is important to prioritize continued vaccinations to maintain optimal health and prevent the spread of preventable diseases. In the past two decades of research and practice, it has been discovered that using the same antigen and delivering it through different vaccine forms, known as sequential immunization or heterologous prime-boost immunization, results in better immunization outcomes than homologous immunization [33,34,35].

Polio vaccines were the first vaccines to be used in humans in a sequential immunization strategy, as many countries immunized against polio with IPV followed by a booster dose of OPV due to economic and production capacity constraints. Although IPV is safer, it has shown a decline in antibody yield compared to the IPV–OPV sequential immunization strategy [36].

Several recent cohort studies on sequential immunization against SARS-CoV-2 have shown that certain combinations of heterologous prime booster immunization strategies including mRNA vaccines, recombinant adenoviral vector vaccines, recombinant protein vaccines, and inactivated vaccines yielded higher gains in humoral and cellular immunity than homologous immunization. For instance, in a study conducted by Chiu and colleagues [37], it was found that when a regimen consisting of both mRNA vaccine and recombinant Ad5 adenovirus vector vaccine was given, the Ad5-mRNA strategy was more likely to produce the highest neutralizing antibody titers, T-cell responses, and cross-neutralizing antibodies against the mutant strains. Furthermore, Li et al. [38] demonstrated that a sequential immunization regimen using inactivated CoronaVac and Ad5 adenovirus vector vaccines was found to be safe, had better immunogenicity, and resulted in a strong elevation of antibody levels in the body compared to full immunization with inactivated vaccines. Results from a study conducted by Zuo et al. [39] indicated that after primary immunization with an inactivated vaccine, booster immunization with an mRNA vaccine significantly increased the level of T- and B-cell responses and enhanced protection against strains including the Omicron variant.

Similar to the findings of previous studies, our study found that heterologous prime-boost immunization resulted in better immunogenicity. Sequential boosting with mRNA or recombinant protein vaccines produced stronger and more persistent humoral immune responses, including binding antibodies, neutralizing antibodies, antibody persistence, and cross-protection against mutant strains. Notably, all four groups experienced a more than ten-fold reduction in neutralizing antibodies against both Delta and Omicron BA.1 variants, indicating that a vaccination containing only the WT strain is insufficient for providing adequate protection against future variants and that immunization with antigens containing various variant strains may be more effective for improving the immune response. Perhaps vaccinating with antigens containing various mutant strains would offer more adequate protection [40,41].

Moreover, our findings indicate that heterologous booster immunization generated more potent antibody-secreting cells and facilitated IgG type switching. The number of antibody-secreting cells noticeably increased after administering the third booster immunization. Antibody-secreting cells are a vital component of adaptive immunity. As terminally differentiated B-cells, they are accountable for the large-scale synthesis and secretion of specific antibodies. This is responsible, in part, for the significant elevation in IgG and neutralizing antibody levels after the third dose of booster immunization. In the antibody typing results, IgG2a was predominant in all groups, which is similar to some of the previously reported literature on the immune bias of the first dose of vaccine determining one of the types of antibodies produced by the body [42]. An inactivated vaccine served as the initial vaccination dose for all groups in this study, specifically the SARS-CoV-2 inactivated vaccine, which induced IgG2a-based antibodies. However, following sequential immunization with a recombinant protein vaccine and mRNA vaccine, we observed an increase in the IgG1/IgG2a ratio from 0.33 to 0.74 and 0.56, respectively, compared to 0.39 with the homologous booster inactivated vaccine.

The germinal center of the lymph node is where B-cells mature and convert antibody types by coming into contact with the antigen to activate and undergo negative and positive selection, affinity maturation, high-frequency somatic cell mutation, and antibody type conversion. This results in the formation of B-cells with a high affinity for antigens that are able to secrete high-affinity antibodies [43,44,45]. Moreover, T follicular helper cells play a crucial role in supplying the activation signals B-cells require, presenting antigens to B-cells in a persistent manner, helping B-cells complete the process of affinity maturation, and regulating the immune response throughout the germinal centers so that the body maintains an appropriate immune response intensity [46]. The results of the present study also demonstrate that the number of GC B-cells and Tfh-cells was higher in groups C and D than in groups A and B. It was also observed that, after heterologous booster immunization, the number of IL-4-secreting cells was significantly increased; IL-4 is a cytokine that promotes the development of Th2-cells and plays a key role in enhancing the humoral immune response. The findings of this study demonstrate that a heterologous primary-boosted immunization strategy improves the immune response in the germinal centers, promotes the development and maturation of B-cells, and promotes the formation of antibody-secreting cells.

At the cellular immunity level, the evidence indicates that boosting immunization, especially heterologous boosting immunization, can significantly enhance the body’s cellular immune response. Groups C and D in this study had a higher number of IFN-γ-secreting cells, a crucial cytokine that enhances cellular immunity and activates dendritic cells, macrophages, and other innate immune cells. It also improves antigen processing and presentation, promoting the formation of a Th1 immune response and CTL-cells, which play vital roles in eliminating pathogen-infected cells for defense purposes [47]. In addition, cytokine analysis in mouse serum confirmed that different immunization strategies do not result in significantly higher abnormalities in cytokine levels at 14 days post immunization, but it can lead to a slight upregulation of factors such as IL-4, IL-5, and IL-10.

Another aspect of this study examined the correlation between enhanced immunity and the evolution of BCR profiles. BCR is an important component of B-cell antigen recognition and signaling consisting of two chains, the heavy chain and the light chain. The creation of distinct and extensive antigen-specific BCR profiles is dependent on a sequence of regulatory procedures of heavy and light chain genes following B-cell exposure to antigens. Important steps include the rearrangement of V(D)J genes, which occurs in response to the action of the RAG enzyme, stochastic splicing, additions or deletions of bases, and somatic high-frequency mutations [48,49]. The extensive rearrangement of the V, D, and J genes through random addition and deletion of bases enables the organism to establish a diverse BCR library, effectively guarding against reinvasion by pathogens. The use of germline genes encoding IgH and IgK in this study uncovered similarities between these and the transcription of B-cell germline genes in naïve mice. For instance, the IgK genes for IGKV4, IGKV3, IGKV1, and IGKV6 were extensively transcribed, and comparatively less so for IGKJ3 and IGKJ4. Mice tend to preferentially transcribe some common germline genes for development and rearrangement after antigenic stimulation. Additionally, the four groups shared some similarities but also exhibited individual differences in the transcription of germline genes, including the previously mentioned IGHV1-77, IGHV3-8, IGHV1-9, and IGHV3-2 in IgH, and IGKV4-59, IGKV4-53, IGKV9-120, and IGKV10-96 in IgK, which were equally present in all groups but underwent an increase or decrease in expression frequency. Furthermore, the four groups demonstrated both commonalities and distinct differences in the utilization of germline genes. Notably, the previously mentioned IGHV1-77, IGHV3-8, IGHV1-9, and IGHV3-2 in IgH, and IGKV4-59, IGKV4-53, IGKV9-120, and IGKV10-96 in IgK were uniformly present across all groups but exhibited either increasing or decreasing variations in frequency. Regarding paired V-J genes, the four groups shared certain high-frequency V-J paired genes while also possessing distinct pairs. This suggests that enhanced immunization fosters the continued development of germline genes after the initial immunization.

In the heavy and light chains of the B-cell receptor, three complementary determining regions (CDR1, CDR2, and CDR3) play crucial roles in recognizing and binding specific antigens. Among these, CDR3 is highly variable and serves as the most crucial complementary determining region for antigen recognition [50]. Hence, the diversity within the CDR3 region contributes additional immune defenses against specific antigens in the adaptive immune response. We also conducted an analysis of the CDR3 sequence of BCR. First, it was noted that all four groups exhibited a greater abundance of H-CDR3 compared to K-CDR3, implying a richer diversity in H-CDR3. We hypothesize that this phenomenon arose from additional D genes participating in gene rearrangements in IgH, coupled with the predominant occurrence of the addition of random nucleotides in IgH. This deduction was made through a comparison of the ratio of the total number of CDR3 species to the number of sequences. When comparing the H-CDR3 and K-CDR3 of groups A, B, C, and D, group A exhibited the highest similarity, with 32.8% and 44.0% of the species being the same, respectively. Group D showed the lowest similarity, followed by group C. Further analysis indicated that the diversity of CDR3 was highest within group D compared to the other three groups. These findings collectively suggest that heterologous boosting immunization can generate a more diverse repertoire of CDR3 species than homologous immunization.

However, this study is limited by a lack of validation for the functional characterization of the complete BCR repertoire through the expression of all monoclonal antibodies. Moreover, the study is based on an animal model, which differs significantly from the human immune system. Finally, we did not compare the strategies of using mRNA and recombinant protein vaccines for initial immunization with inactivated vaccines for booster immunization. It is not possible to determine whether these prime–booster immunization strategies are inferior to the immunization strategies discussed in this paper.

Furthermore, it is important to emphasize that the vaccines referred to in the text are those described in the material. For example, the mRNA vaccines described in the text are not BNT162b2 and mRNA1273, which are two vaccines used widely across the world.

## 5. Conclusions

The mechanism behind the sequential immunization strategy of vaccines is a promising area for our research. The current study offers insight into the mechanism of sequential immunization in developing the BCR repertoire.

Based on our results, we conclude that through heterologous booster immunization based on inactivated vaccination, significant improvements were observed in both humoral and cellular immune responses to SARS-CoV-2 compared to homologous booster. These included significant enhancement in neutralizing and binding antibodies, cross-protection against mutant strains, and an increase in the number of IFN-γ-secreting cells. During the formation of B-cell receptor lineages that specifically target a virus, the inactivated vaccine provides an initial foundation for the organism. The development of the BCR lineage occurs in a graded fashion. After homologous and heterologous booster immunizations, the previous BCR lineage retains some high-frequency germline genes. Additionally, a unique BCR lineage is formed through new V(D)J rearrangement and other mechanisms. Each individual has a distinct BCR profile, which becomes more diverse over time.

## Figures and Tables

**Figure 1 vaccines-12-00120-f001:**
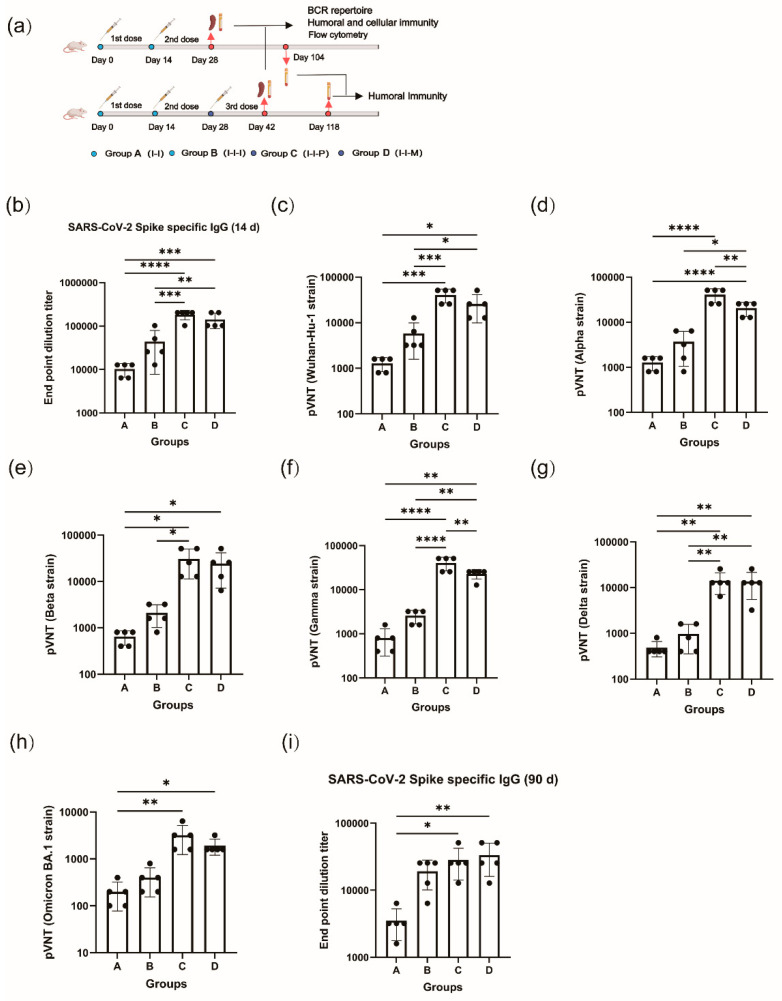
Booster immunization significantly increases antigen-specific antibodies in mice. (**a**) Immunization procedures for mice; Group A: 2×INA (I-I), Group B: 3×INA (I-I-I), Group C: 2×INA+protein (I-I-P), Group D: 2×INA+mRNA (I-I-M). (**b**) SARS-CoV-2 S1-specific total IgG levels were assessed 14 days post immunization in four groups of mice. (**c**–**h**) The neutralizing antibody titers of pseudovirus were analyzed for six different variants of the virus, including WT, Alpha, Beta, Gamma, Delta, and Omicron BA.1. (**c**) WT, (**d**) Alpha, (**e**) Beta, (**f**) Gamma, (**g**) Delta, (**h**) Omicron BA.1. (**i**) SARS-CoV-2 S1-specific total IgG titers 90 days after immunization. * *p* < 0.05, ** *p* < 0.01, *** *p* < 0.001, **** *p* < 0.0001.

**Figure 2 vaccines-12-00120-f002:**
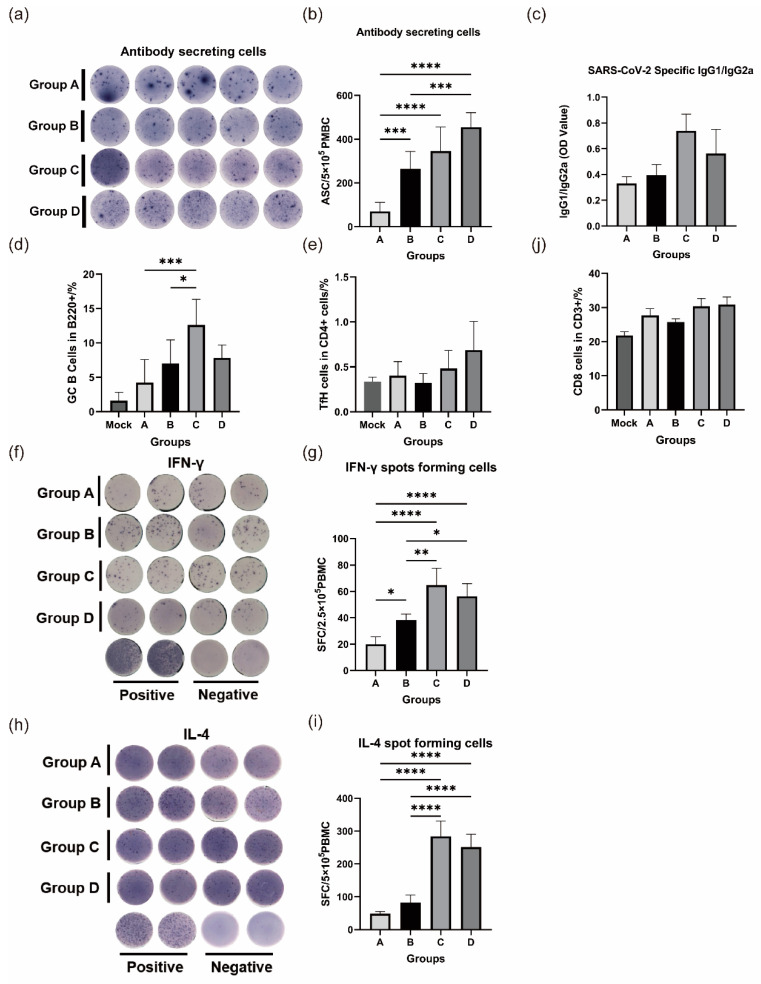
Heterologous booster immunization elicits stronger antibody-secreting cells and induces IgG antibody class switching while enhancing the T-cell response (Group A: I-I; Group B: I-I-I; Group C: I-I-P; Group D: I-I-M). (**a**,**b**) Splenic lymphocytes from four groups of mice were analyzed for their ability to produce ASCs in response to the S1 antigen. (**c**) Ratios of IgG1 to IgG2a in the four groups of mice. (**d**,**e**) GC center reaction. (**d**) Proportions of GC B-cells in lymph nodes. (**e**) Proportions of TFH cells in lymph nodes. (**f**–**j**) T-cell response of four groups of mice. (**f**,**g**) S1-specific IFN-γ-secreting cells in splenic lymphocytes. (**h**,**j**) S1-specific IL-4-secreting cells in splenic lymphocytes. (**j**) Proportion of CD8+ splenic lymphocytes.). * *p* < 0.05, ** *p* < 0.01, *** *p* < 0.001, **** *p* < 0.0001.

**Figure 3 vaccines-12-00120-f003:**
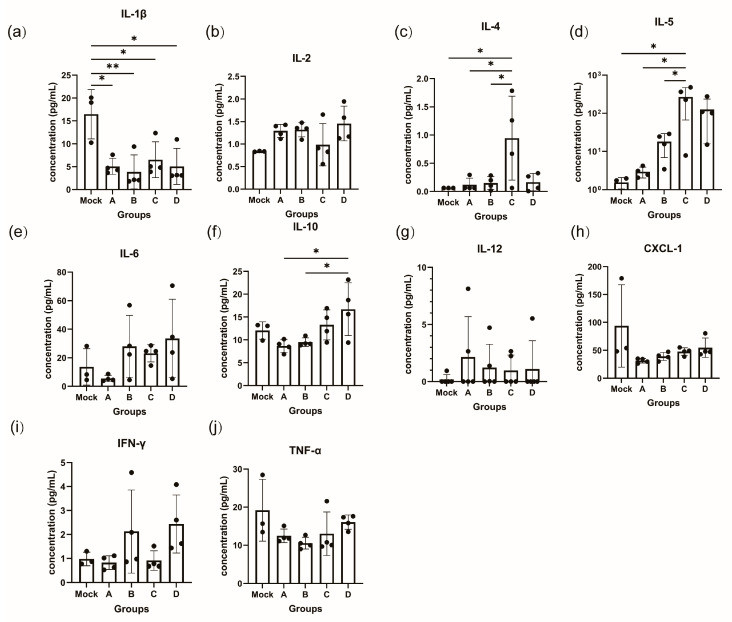
Concentration of ten cytokines in the serum of mice sacrificed at the completion of the immunization schedule (Group A: I-I; Group B: I-I-I; Group C: I-I-P; Group D: I-I-M). (**a**) IL-1β, (**b**) IL-2, (**c**) IL-4, (**d**) IL-5, (**e**) IL-6, (**f**) IL-10, (**g**) IL-12, (**h**) CXCL-1, (**i**) IFN-γ, (**j**) TNF-α. * *p* < 0.05, ** *p* < 0.01.

**Figure 4 vaccines-12-00120-f004:**
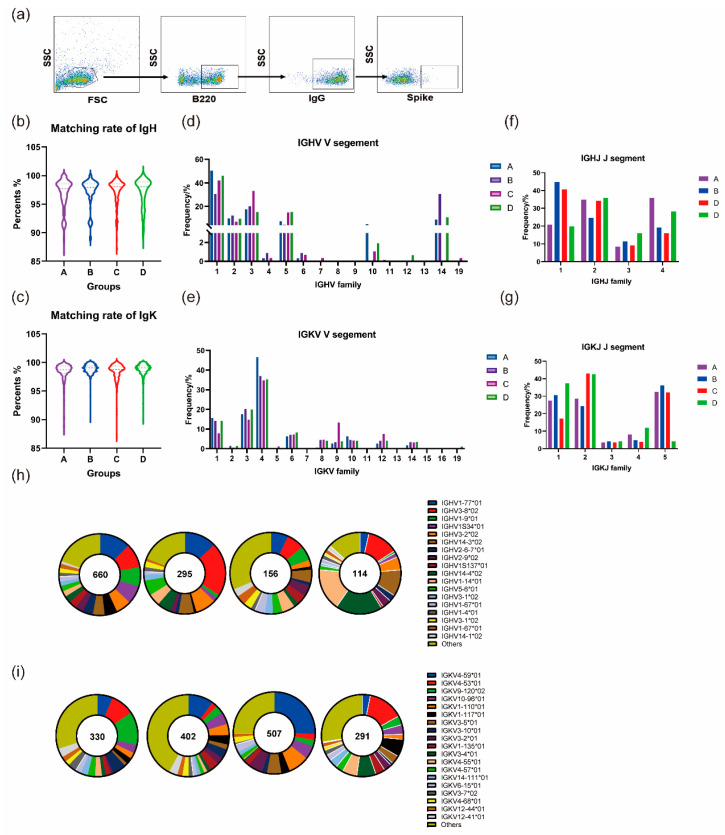
Flow cytometric sorting of antigen-specific B-cells and analysis of lineage distribution (Group A: I-I; Group B: I-I-I; Group C: I-I-P; Group D: I-I-M). (**a**) Flow cytometry sorting of antigen-specific B-cells. (**b**,**c**) Somatic hypermutation in IgH- and IgK-encoding genes. (**d**–**f**) Distribution of germline genes encoding IgH. (**e**–**g**) Distribution of germline genes in IgK. (**h**,**i**) Distribution of germline high-frequency gene subclasses encoding IgH and IgK.

**Figure 5 vaccines-12-00120-f005:**
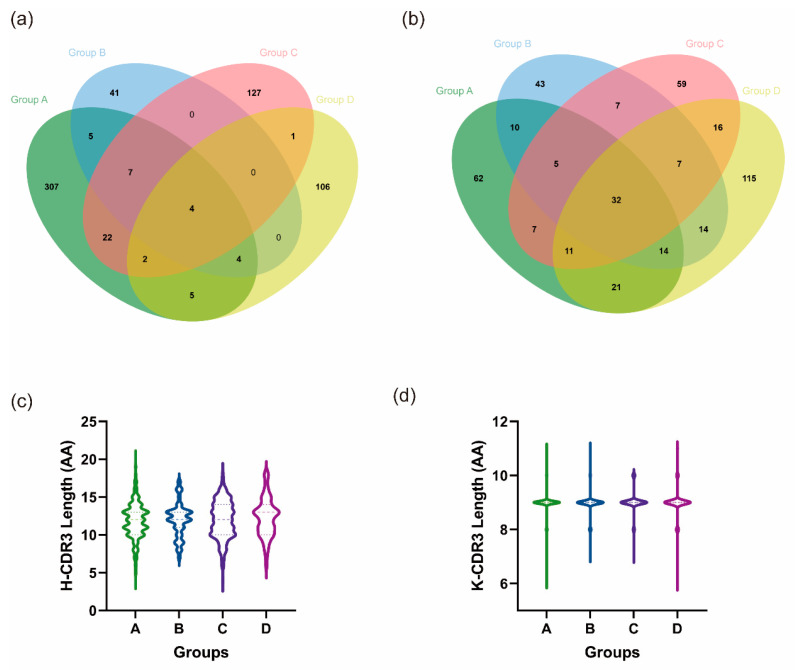
Heterologous boosting increases the diversity of CDR3 (Group A: I-I; Group B: I-I-I; Group C: I-I-P; Group D: I-I-M). (**a**,**b**) Number of common and unique CDR3 sequences in the four groups. (**a**) H-CDR3. (**b**) K-CDR3. Green ovals represent group A, blue ovals represent group B, red ovals represent group C, and yellow ovals represent group D. The overlap between them indicates the number of shared sequences. (**c**,**d**) Length distribution of CDR3. (**c**) H-CDR3. (**d**) K-CDR3.

**Table 1 vaccines-12-00120-t001:** Number of sequences and CDR3 species obtained.

Group	Sequence Number of IgH	Species of H-CDR3	Ratio	Sequence Number of IgK	Species of K-CDR3	Ratio
A	660	355	0.54	507	162	0.32
B	114	61	0.54	291	132	0.45
C	295	163	0.55	328	144	0.44
D	156	123	0.79	402	230	0.57

## Data Availability

Data are contained within the article and Appendix A.

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
