# Peer review of "Heterologous Booster Immunization Based on Inactivated SARS-CoV-2 Vaccine Enhances Humoral Immunity and Promotes BCR Repertoire Development"

_vaccines, 2024, doi:10.3390/vaccines12020120_

Round 1

Reviewer 1 Report

Comments and Suggestions for Authors

Interesting manuscript on the use of different vaccines for booster doses in vaccination schedules and their effect on the immune response, using COVID-19 as a model and considering various vaccine technologies available. In general, the text is clear and well-written, with a well-placed experimental strategy. There are typographical errors, such as using "exponential" for cell counting and subscript for scientific notation of carbon dioxide.

My critiques of the work itself were highlighted as weaknesses in the last paragraph of the discussion: the use of a murine model to characterize the immune response of an event already described in clinical trials in humans. I emphasize here that the novelty lies in the characterization of BCRs. 

In conclusion, despite these flaws, I understand that the article has value and deserves to be published after the minor corrections pointed out.

Reviewer 2 Report

Comments and Suggestions for Authors

heterologous schedule seem to have a more profound effect on GC, due to the effectiveness of mRNA vaccines and the authors studied some features of the response in a mouse model . SARS-2, differently from other infections, is a "moving target" due to the emergence of VOCs over time.

- precise the "serious coagulation problems" are mostly VITT, a very rare event. the problem was most the low adherence to this vaccine type, due to multiple reasons

- the authors should stress that their results are obtaind with products different from the more used BNT162b2 and derivations, MRNA1273

Comments on the Quality of English Language

improve the flow of sentences

Reviewer 3 Report

Comments and Suggestions for Authors

The authors compare the immunogenicity of a third dose of inactivated SARS-CoV-2 virus (ancestral) produced by the chinese Academy of Medical Sciences, a recombinant spike protein (ancestral) produced by SinoBiological adjuvanted with AddaVax, and an mRNA vaccine provided from Stemirna Co.

The article emphasizes the importance of a heterologous booster, however it overlooks a more important variable which is the apparent lower immunogenicity of the inactivated vaccine compared to the other two vaccines used. To be able to conclude that is the heterologous combination what improves the immunogenicity and not the higher immunogenicity of the protein or the mRNA vaccine, mice vaccinated with the recombinant protein, or the mRNA followed by the inactivated vaccine should be evaluated.  If the authors cannot provide this data, please then rewrite the article accordingly.

Please provide further description of the three vaccines used. For example, provide the typo of mRNA vaccine and whether lipid nanoparticles were used. For the protein subunit, provide the catalogue number (if any), or the sequence, trimerization domain (if any).

Why does Mock animals have higher cytokine responses than the immunized mice? If mock means unvaccinated animals, and given the huge variability in just that group, how are the responses seen real and relevant, specially given the longer time point after vaccination?

In regards to the CDR3 species, are these differences statistically significant? What would this diversity mean? Will these animals have more cross-reactive antibodies, of so from the pVNT data it only looks like that the more immunogenic booster, increases the pVNT proportionally against the different strains tested.

Please include in the legend of each figure the description of each vaccination.

Reviewer 4 Report

Comments and Suggestions for Authors

The paper compares the immune response elicited in mice by different vaccination procedures.

The data indicate that heterologous boosting induces higher titer of antibodies, higher protection, more robust cellular responses, greater number of ASC and more extensive diversification of B cell clones.

The experimental approach is adequate, the data are solid, well presented and correctly discussed.

No modification is required

Author Response

Dear Reviewer,

The co-anthors and I would like to thank you for the time and effort spent in reviewing the manuscript. Thank you for your approval of our work!